# Genome-Wide Identification of SPX Family Genes and Functional Characterization of *PeSPX6* and *PeSPX-MFS2* in Response to Low Phosphorus in *Phyllostachys edulis*

**DOI:** 10.3390/plants12071496

**Published:** 2023-03-29

**Authors:** Jiali Luo, Zhihui Liu, Jiawen Yan, Wenhui Shi, Yeqing Ying

**Affiliations:** State Key Laboratory of Subtropical Silviculture, Zhejiang A&F University, Hangzhou 311300, China

**Keywords:** *Phyllostachys edulis*, SPX, low phosphorus, genome-wide analysis

## Abstract

Moso bamboo (*Phyllostachys edulis*) is the most widely distributed bamboo species in the subtropical regions of China. Due to the fast-growing characteristics of *P. edulis*, its growth requires high nutrients, including phosphorus. Previous studies have shown that SPX proteins play key roles in phosphorus signaling and homeostasis. However, the systematic identification, molecular characterization, and functional characterization of the SPX gene family have rarely been reported in *P. edulis*. In this study, 23 SPXs were identified and phylogenetic analysis showed that they were classified into three groups and distributed on 13 chromosomes. The analysis of conserved domains indicated that there was a high similarity between PeSPXs among SPX proteins in other species. RNA sequencing and qRT-PCR analysis indicated that *PeSPX6* and *PeSPX-MFS2*, which were highly expressed in roots, were clearly upregulated under low phosphorus. Co-expression network analysis and a dual luciferase experiment in tobacco showed that PeWRKY6 positively regulated the *PeSPX6* expression, while PeCIGR1-2, PeMYB20, PeWRKY6, and PeWRKY53 positively regulated the *PeSPX-MFS2* expression. Overall, these results provide a basis for the identification of SPX genes in *P. edulis* and further exploration of their functions in mediating low phosphorus responses.

## 1. Introduction

Moso bamboo (*Phyllostachys edulis*) is the most widely distributed bamboo species in the subtropical regions of China. The planting area is 4,677,800 hectares, accounting for 70% of the total bamboo forest area in China [1,2,3]. *P. edulis* not only has the highest ecological and economic value, but also has the highest cultural value of all bamboos. Due to the fast-growing characteristics of *P. edulis*, its growth requires a high amount of nutrients [4]. Thus, an adequate supply of soil nutrients is necessary to ensure the normal growth of *P. edulis*.

As an essential mineral element during plant growth and development, phosphorus (P) plays key roles in energy transmission, signal transduction, photosynthesis, and respiration [5,6,7]. Plants prefer to obtain inorganic soluble phosphate in the soil through the root system, and cannot directly assimilate the other forms of P, such as organic phosphate (Pi) [8]. However, the content of inorganic phosphorus in soil is very low, and it does not diffuse easily in the soil. As a result, plants have evolved mechanisms to adapt to Pi-deficient conditions [9]. The phosphorus starvation signaling pathway in plants has been well documented [10].

Among them, SPX proteins play very important roles in P signaling and homeostasis [11,12]. The SPX domain is a conservative domain named after SYG1 (suppressor of yeast gpa1), Pho81 (CDK inhibitor in yeast PHO pathway), and XPR1 (xenotropic and polytropic retrovirus receptor). Based on the presence of additional domains, SPX proteins can be further classified into four subfamilies: SPX proteins, which contain only an SPX domain; SPX-EXS proteins, which have an SPX domain and an EXS (ERD1, XPR1, and SYG1) domain; SPX-MFS proteins, which have an SPX domain and a major facility superfamily (MFS) domain; and SPX-RING proteins, which contain an SPX domain and a RING-type zinc finger domain [13,14,15]. SPX proteins have been identified in many plant species, including *Arabidopsis*, rice, and maize [16]. For example, there are 20 SPX domain-containing proteins in *Arabidopsis*. *AtSPX1* participates in the transcription regulation of Pi-responsive genes, while *AtSPX3* functions in the potential negative feedback regulation of the P signal network [17]. 33 SPX gene family members have been found in maize. Xiao et al. [18] found that ZmSPX4.1 and ZmSPX4.2 showed strong responses to low Pi stress and exhibited remarkably different expression patterns in low-Pi sensitive and insensitive cultivars of maize [18]. Currently, the role of SPX genes in response to low phosphorus stress in *P. edulis* is still poorly understood. Thus, the systematic identification, molecular characterization, and functional characterization of the SPX gene family in response to low Pi in *P. edulis* are of great importance to improve the absorption and utilization of Pi in *P. edulis*.

In this study, the whole genome of SPX members in *P. edulis* was first identified and characterized. The chromosomal mapping, phylogenetic relationships, and conserved motifs of *SPX* genes were analyzed. Then, the expression levels of SPX members exposed to low Pi were analyzed using RNA sequencing (RNA-seq) and quantitative real-time PCR (qRT-PCR). Furthermore, several transcriptional factors were identified that modulated the expression of *PeSPX6* and *PeSPX-MFS2* in *P. edulis*.

## 2. Result

### 2.1. Identification of SPX-Domain-Containing Proteins in P. edulis

To identify *P. edulis* SPXs, the conserved PF03105 domain was used as the probe to execute a genome-wide search of candidate genes with the HMMER tool, and a total of 30 putative protein hits were obtained. To verify the HMMER search results, domain analysis was further performed for the 30 putative proteins with CDD and SMART. Among the 30 putative proteins, 23 harboring the conserved SPX domains were identified using both CDD and SMART. These proteins were designated PeSPX1 through PeSPX1-EXS6 and were considered authentic SPX candidates in the *P. edulis* genome. The length of these protein-coding regions ranged from 738 bp to 2571 bp. These proteins consisted of 245–856 amino acids, and the corresponding molecular weights ranged from 27.97 to 97.55 kDa. The isoelectric point (pI) value of these PeSPX proteins ranged from 5.04 to 9.4 (Table 1).

The chromosomal distribution of the *P. edulis* SPX-domain-containing proteins genes is illustrated in Figure 1. Furthermore, 23 PeSPX genes were distributed on 13 *P. edulis* chromosomes. Chromosomes 4, 5, 10, 14, 20, 23, and 24 each contained one gene; chr 3, 15, and 21 each had two genes; chr 8 and 17 each had three genes; and chr 6 had four genes (Figure 1).

### 2.2. Phylogenetic Analysis of the SPX-Domain-Containing Protein Genes in P. edulis and Other Species

To evaluate the evolutionary relationships of SPX-domain-containing protein genes in *P. edulis*, this work analyzed the sequence features in three different species, including *A. thaliana*, *O. sativa*, and *P. edulis*, and a total of 56 SPX-domain-containing protein genes were used to construct a phylogenetic tree with the neighbor-joining (NJ) method using MEGA (version 7) (Figure 2). The phylogenetic tree indicated that PeSPXs could be divided into three subfamilies (the SPX, SPX-MFS, and SPX-EXS subfamilies). In addition, the numbers of SPX subfamily proteins in the three species were highly asymmetrical. For example, 11 PeSPXs, five OsSPXs, and four AtSPXs were classified in the SPX subfamilies, and six PeSPXs, 11 AtSPXs, and three OsSPXs were included in the SPX-EXS subfamilies. Only one OsSPX and two AtSPXs were classified in the SPX-RING subfamilies. These results indicate that the SPX gene family is highly conserved and diverse in different plants.

### 2.3. Structure Analysis of the SPX-Domain-Containing Genes and Proteins in P. edulis

All SPX-domain-containing proteins contain the SPX domain in the C-terminal portion (Figure 3). The three subfamilies of SPX-domain-containing proteins (SPX, SPX-EXS, and SPX-MFS) were found in *P. edulis*. The SPX subfamily possessed 11 members, the SPX-EXS subfamily contained six members, and the SPX-MFS subfamily had six members.

Structural features were then characterized for PeSPX genes, and a large divergence in exon number was observed: three exons were detected for PeSPX subfamily, ten for PeSPX-MFS subfamily, 11 for *PeSPX-EXS1*, 13 for *PeSPX-EXS4*, 14 for *PeSPX-EXS2*, *PeSPX-EXS3*, *PeSPX-EXS5*, 15 for *PeSPX-EXS6* (Figure 3A)*. PeSPX-MFS3* had the greatest length (approximately 31 kb), with only 10 exons and nine introns.

### 2.4. Expression Analysis of P. edulis SPX-Domain-Containing Protein Genes

To explore the effects of low phosphorus on the growth and development of *P. edulis*, *P. edulis* plants were treated with 1/2 Kimura nutrient solution in different Pi levels. Under low phosphorus conditions, the root length was significantly lower than that under the normal Pi supply, but the number of lateral roots was significantly higher than that under the normal Pi supply (Figure 4A,B). Furthermore, the phosphorus content in the shoots and roots of *P. edulis* was also measured. As shown in Figure 4C, the phosphorus content in both shoots and roots of *P. edulis* treated with low phosphorus was much lower compared to those in shoots and roots of *P. edulis* with a normal phosphorus supply. To further explore the molecular mechanism of *P. edulis* in response to low phosphorus, RNA-seq was performed (Appendix A). The expression data of 23 *PeSPX*s were clustered and displayed in a heat map (Figure 4). The expression levels of five genes were upregulated, while 18 genes were downregulated under low phosphorus stress.

Next, two *PeSPXs* (*PeSPX6* and *PeSPX-MFS2*) with higher expression levels were selected for qRT-PCR analysis. Consistent with the RNA-seq result, the expression levels of *PeSPX6* and *PeSPX-MFS2* were clearly upregulated under low phosphorus (Figure 5A). Furthermore, the expression patterns of *PeSPX6* and *PeSPX-MFS2* were also analyzed. As shown in Figure 5B, *PeSPX6* and *PeSPX-MFS2* were more highly expressed in the roots, while the expression was lower in the leaves and stems of *P. edulis*.

### 2.5. Validation of SPX Gene Network Regulation

Because the expression levels of *PeSPX6* and *PeSPX-MFS2* were increased by low phosphorus, this study then investigated what transcriptional regulation of *PeSPX6* and *PeSPX-MFS2* might exist in response to low phosphorus. Pearson’s correlations were performed between the expression levels of *PeSPX6* and *PeSPX-MFS2* and transcription factors induced by low phosphorus (Figure 6, Appendix A). Eight transcription factors with high expression levels under low phosphorus stress were selected to explore their function in regulating the expression levels of *PeSPX6* and *PeSPX-MFS2* using a dual-luciferase analysis (Appendix A). The promoters of *PeSPX6* (2000 bp) and *PeSPX-MFS2* (2000 bp) were separately cloned and fused to the firefly luciferase protein (Fluc) at the N-terminus, which also had a Renilla luciferase (Rluc). The Fluc/Rluc ratio represents the ability of transcription factors to transcriptionally activate the *PeSPX6* or *PeSPX-MFS2* promoter. As shown in Figure 6, PeWRKY6 enhanced the activity of luciferase driven by the *PeSPX6* promoter, while PeCIGR1-2, PeMYB20, PeWRKY6, and PeWRKY53 enhanced the activity of luciferase driven by the *PeSPX-MFS2* promoter.

## 3. Discussion

SPX family genes are widely found in eukaryotes, including plants, fungi, and multicellular animals [19,20]. At present, the SPX family genes have been extensively studied in *Arabidopsis* [17] and rice [21,22]. In rice, there are six SPX subfamily genes. Studies on OsSPX1 and OsSPX genes in rice have found that these two genes can participate in the negative regulation of PHR2 under the condition of rich phosphorus [21]. The SPX-EXS family, involved in phosphorus transport and signaling from root to terminal bud in rice and Arabidopsis thaliana [23,24,25]. Members of the SPX family, including 20 AtSPXs, 69 BnaSPXs, 9 GmSPXs, 14 OsSPXs, and 46 TaSPXs, have been reported and characterized by bioinformatics analysis [17,19,26,27]. However, the functional analysis of the SPX family in *P. edulis* has not been reported. In the present study, systematic identification, molecular characterization, and functional characterization of the SPX gene family were performed.

Previous studies have shown that SPX family genes play an important role in the sensing, signal transduction, and transport of inorganic phosphate (Pi) in eukaryotes, such as PHO1 (SPX-EXS) [28], PHT5 (SPX-MFS) [29] and AtNLA (SPX-RING) [30]. *AtSPX1* was expressed most rapidly in Pi starvation induction, which indicated that *AtSPX1* has a potential transcriptional regulation effect on Pi starvation [19]. In rice, eight SPX genes (*OsSPX1, OsPHO2*) were significantly increased under exposure to low phosphorus [19,31]. The expression of *TaSPX2* was significantly induced in wheat under low P stress [32]. It was found that *SPX2* was induced to express in tea plants under low phosphorus stress [33]. Bn SPX2 was continuously induced to express in various tissue parts of Brassica napus under low P stress [27]. In the present study, only five SPX genes were clearly upregulated under low phosphorus. Among them, two genes (*PeSPX6* and *PeSPX-MFS2*) showed higher expression levels compared to the others in the roots. This strongly implies that *PeSPX6* and *PeSPX-MFS2* might regulate the low phosphorus response in *P. edulis*. In addition, the tissue-specific expression indicated that *PeSPX6* and *PeSPX-MFS2* were more highly expressed in roots, which further supported the hypothesis that *PeSPX6* and *PeSPX-MFS2* were involved in phosphorus nutrient uptake in the roots of *P. edulis*. In future research, the biological function of these genes in *P. edulis* needs to be explored.

Increasing evidence shows that proteins containing the SPX domain are key players in a series of processes that control the dynamic balance of phosphorus in plants. In *Arabidopsis*, the physical interaction between AtSPXs and AtPHRs under sufficient Pi conditions prevents AtPHRs from binding to the promoter of *AtPSI*, thus inhibiting the PHR transcriptional activity [34,35,36]. *AtSPX3* is involved in responses to low Pi stress [23]. In rice, phosphorus starvation induces the accumulation of OsSPX3 to restore the phosphorus balance [37]. *PeSPX6* has the highest similarity to *OsSPX3* and *AtSPX3*, suggesting that *PeSPX6* might be involved in the response to low phosphorus and play a role in the Pi starvation signal transduction pathway of *P. edulis*. As a volatile phosphate efflux transporter, OsSPX-MFS3 is involved in maintaining phosphate homeostasis in rice. Phylogenetic analysis showed that *PeSPX-MFS2* and *OsSPX-MFS3* had high homology. Thus, this study speculates that *PeSPX-MFS2* might be a low-affinity Pi transporter that mediates Pi efflux from the vacuole into the cytosol. In general, the plant SPX family can be divided into four subclasses based on the presence of additional protein domains, namely SPX, SPX⁃EXS, SPX⁃MFS, and SPX⁃RING [13]. Among them, SPX-RING is the smallest subfamily in the SPX family. For example, there is only one member of the SPX-RING subfamily in rice and tow in *Arabidopsis*. Surprisingly, this subfamily was not found in the *P. edulis* genome. This demonstrates the complexity of the SPX family in different plant species.

Wang et al. [35] showed that *OsSPX1* and *OsSPX2* inhibited phosphate starvation in rice by interacting with PHR2 in a phosphate-dependent manner. AtSPX4 regulates the PHR1-dependent and independent regulation of stem phosphorus status in Arabidopsis [38]. In this study, several transcription factors, namely PeMYB20, PeMYB30, PeWRKY6, PeWYKY53, PeERF110, PeNAC030, PeCIGR1-1, and PeCIGR1-2, were upregulated up by low phosphorus. Furthermore, the dual luciferase experiment in tobacco showed that PeWRKY6 positively regulated the expression of *PeSPX6*, while PeCIGR1-2, PeMYB20, PeWRKY6, and PeWRKY53 positively regulated the expression of *PeSPX-MFS2*. These results imply that these transcription factors might also play a vital role in the adaptation of *P. edulis* to low-phosphorus environments.

In this study, phylogenetic analysis showed that 23 SPXs were classified into three groups and distributed on 13 chromosomes. The analysis of conserved domains indicated that there was a high similarity between PeSPXs among SPX proteins in other species. *PeSPX6* and *PeSPX-MFS2*, which were highly expressed in roots, were obviously upregulated under low phosphorus. PeWRKY6 positively regulated the expression of *PeSPX6*, while PeCIGR1-2, PeMYB20, PeWRKY6, and PeWRKY53 *p*ositively regulated the expression of *PeSPX-MFS2*. Based on this, it is speculated that these genes play different roles in various biological processes, laying a theoretical foundation for elucidating the functions of the SPX family genes in *P. edulis*.

## 4. Materials and Methods

### 4.1. Identification and Classification of SPX Genes

The genomic information of *P. edulis* was downloaded from the Gigascience database (accession numbers: PRJEB2955 and PRJEB2956). Then, using “hmmsearch” with an expected value (e-value) threshold of 0.5 × 10^−3^, the genomic protein sequences of *P. edulis* were searched for the Hidden Markov Model (HMM) profiles of the SPX domain (PF03105). Using the NCBI Conserved Domain Database (NCBI-CDD) (http://www.ncbi.nlm.nih.gov/cdd (accessed on 12 November 2022)) and the SMART database (http://smart.embl-heidelberg.de (accessed on 12 November 2022)) the outcomes were further validated (e-value = 1 × 10^−2^). The SPX protein sequences of Arabidopsis thaliana and Oryza sativa were searched using Phytozome 13 (https://phytozome-next.jgi.doe.gov/ (accessed on 12 November 2022)).

### 4.2. Phylogenetic Analysis of SPX Proteins

Multiple amino acid sequence alignments were performed using MAFFT with the default parameters utilizing *A. thaliana*, *O. sativa*, and *P. edulis* sequences to examine the evolutionary relationships and the full-length protein sequences of *P. edulis* SPXs. Then, using the auto mode in MEGA (version 7), a phylogenetic tree of amino acid sequences based on the neighbor-joining (NJ) was created.

### 4.3. Gene Structure and Chromosomal Location

Conserved motif structures were examined using the MEME online software (meme.nbcr.net/meme (accessed on 12 November 2022)) and the motif function was explored using NCBI-CDD. To display the locations of PeSPXs on chromosomes, TBtools was employed. The genomic sequence information applied to the analysis of both gene structure and chromosomal location was derived from NCBI [39].

### 4.4. Transcriptome Sequencing

High-quality RNA was used to create the cDNA library and the 150-bp paired-end reads that resulted were sequenced using the Illumina NovaSeq 6000 platform. After filtering, the cleaning data were compared to the reference genome of *P. edulis* through HISAT2, and the gene expression levels were counted as Fragments Per Kilobase per Million (FPKM) using StringTie. Based on *p*-values < 0.05 and fold changes ≥ 2, the R package Deseq2 identified differentially expressed genes (DEGs) in each sample group.

### 4.5. Determination of Phosphorus Content

The content of phosphorus in each organ of *P. edulis* was determined using the Mo-Sb colorimetric method [40]. The extracted solution was mixed with 2,4-dinitrophenol, NaOH, and 0.5 mol/L H_2_SO_4_ until the yellow was arised. Then, the mixed solution was reacted with Mo-Sb-Vc chromogenic agent at 25 °C for 30 min. The absorbance at 700 nm was measured using an ultraviolet-visible spectrophotometer (V 5600, Shang Hai, China), then the phosphorus content was calculated.

### 4.6. Real-Time RT-PCR Analysis

Utilizing the cDNA reverse transcription kit, whole pure RNA was transformed into cDNA (PrimeScriptTM RT Master Mix, Takara, Kusatsu, Japan). Then, qRT-PCR was carried out using the CFX96 TouchTM Thermal Cycler and the ChamQ SYBR qPCR Master Mix kit from Vazyme (Vazyme, Nanjing, China) (Bio-Rad, Hercules, CA, USA). The 2^−ΔΔCT^ method was utilized to normalize for transcript levels using the housekeeping gene PeTIP41 (Appendix A) [41].

### 4.7. Luciferase Assays

The full length of transcription factors were separately cloned into pGreenII 62-SK vector. The promoter of *PeSPX6* and *PeSPX-MFS2* (2000 bp) was cloned into pGreenII 0800 LUC vector (Appendix A). *Agrobacterium tumefaciens* GV3101 was used to convert the aforementioned vectors into four-week-old *Nicotiana benthamiana* leaves [42]. After infiltration for 4 days, the activity of LUC and REN were assessed in accordance with the instructions provided in the Dual Luciferase Reporter Gene Assay Kit (Beyotime, Nantong, China).

## Figures and Tables

**Figure 1 plants-12-01496-f001:**
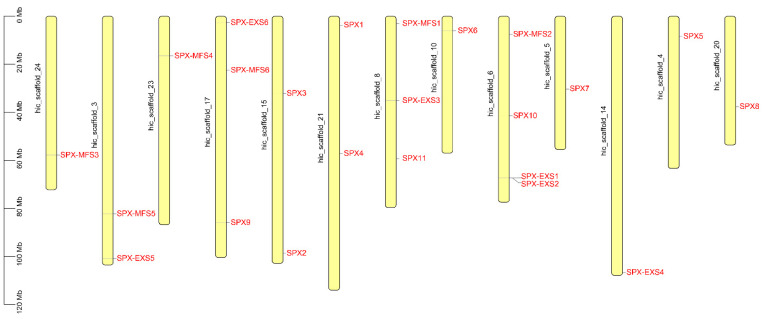
Chromosomal locations of the SPX-domain-containing proteins family genes in *Phyllostachys edulis*.

**Figure 2 plants-12-01496-f002:**
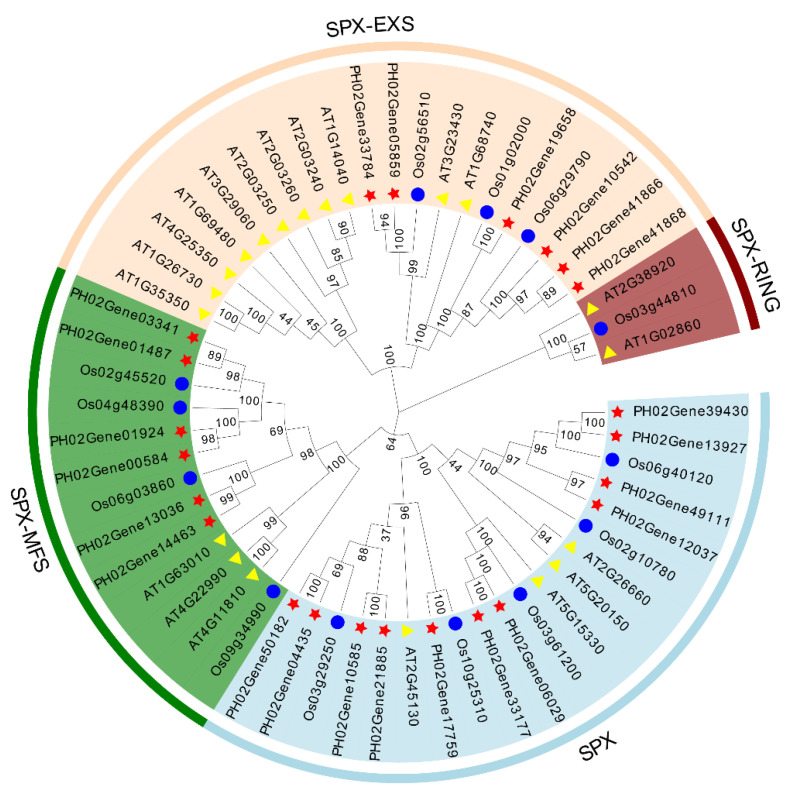
Phylogenetic analysis of the SPX-domain-containing protein genes in *Phyllostachys edulis*, *Arabidopsis thaliana*, and *Oryza sativa*. The phylogenetic tree was constructed with the neighbor-joining (NJ) algorithm and 1000 bootstrap replicates. Different colors indicate the different subfamilies of SPX-domain-containing proteins.

**Figure 3 plants-12-01496-f003:**
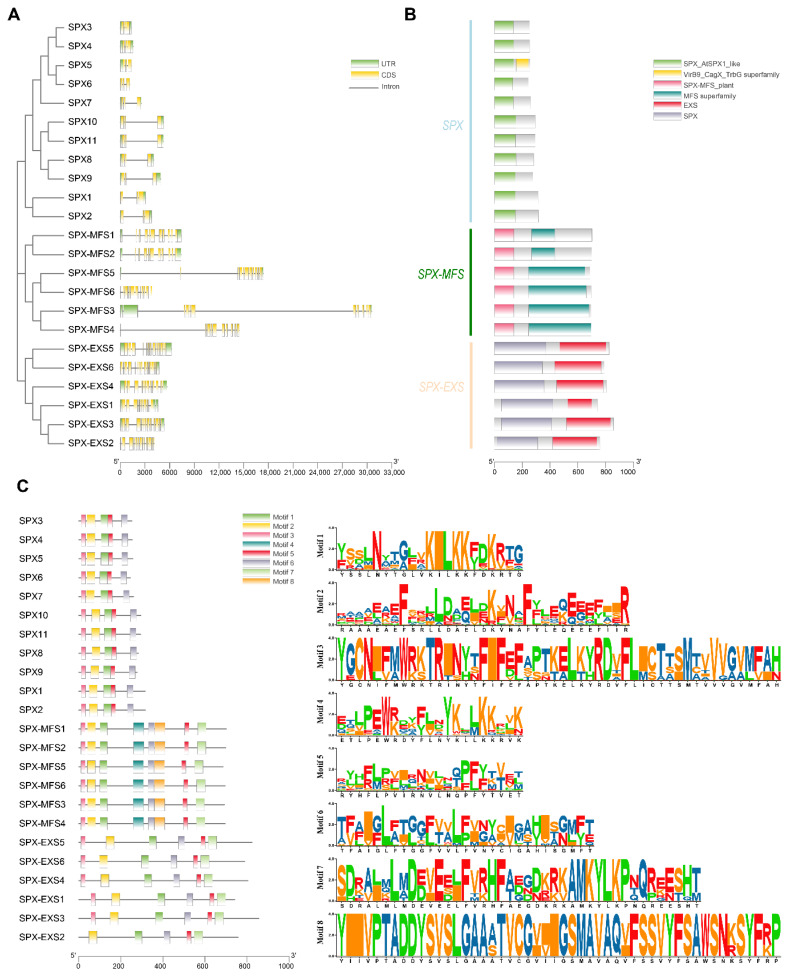
Gene structure, motifs, and conserved structural domains analysis of the PeSPXs in *Phyllostachys edulis*. (**A**) Intron-exon structure of the PeSPX genes. Yellow boxes, black lines, and green boxes represent exons (CDS), introns, and the 5′ and 3′ untranslated regions, respectively. (**B**) Predictions of the conserved domain in 23 PeSPX proteins. The length of each protein sequence is represented by the gray bars, and colored boxes represent conserved domains. (**C**) Distribution of the conserved motifs in PeSPX proteins. The scale bar at the bottom indicates the protein lengths, and sequence logos for each conserved motif are shown on the right. Visualized by TBtools (v1.108) (https://github.com/CJ-Chen/TBtools/releases (accessed on 12 November 2022)).

**Figure 4 plants-12-01496-f004:**
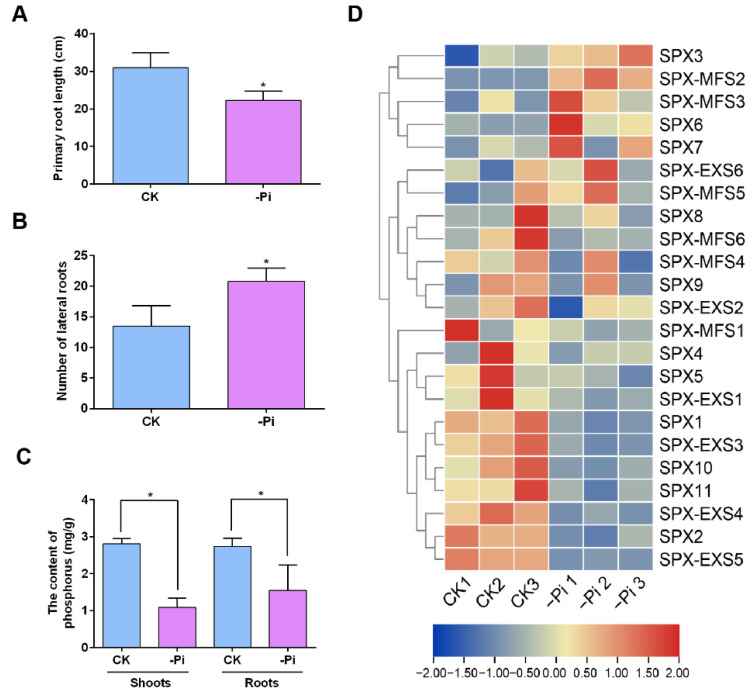
Expression profile of *PeSPXs* exposed to low phosphorus. (**A**) Primary root length of *Phyllostachys edulis* seedlings exposed to low phosphorus. (**B**) Number of lateral roots of *P. edulis* seedlings exposed to low phosphorus. (**C**) Phosphorus content in *P. edulis* roots and shoots exposed to low phosphorus. (**D**) Heatmap of *PeSPXs* exposed to low phosphorus via transcriptome analysis. Error bars in (**A**–**C**) indicate SD (*n* = 3). The asterisk shows a significant difference compared to the control using the unpaired Student’s *t* test (* *p* < 0.05).

**Figure 5 plants-12-01496-f005:**
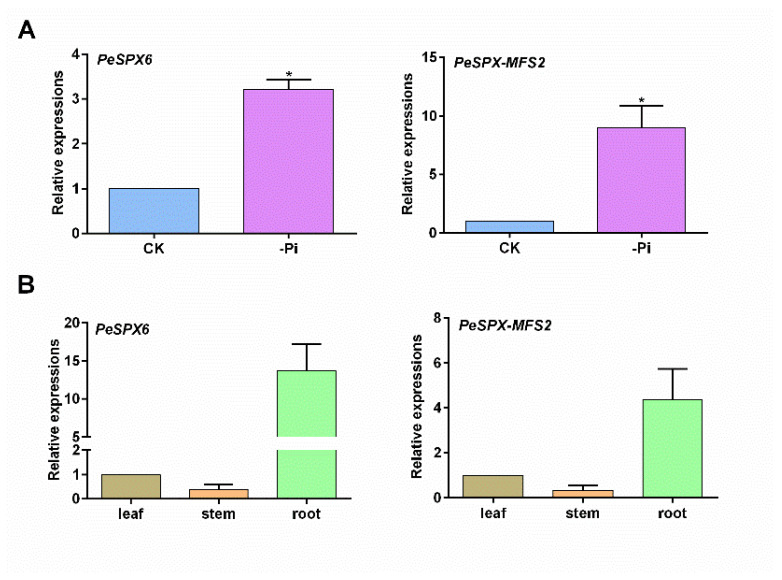
Expression patterns of *PeSPX6* and *PeSPX-MFS2* in *Phyllostachys edulis***.** (**A**) Quantitative RT-PCR analysis of *PeSPX6* and *PeSPX-MFS2* exposed to low phosphorus in roots. (**B**) Quantitative RT-PCR analysis of *PeSPX6* and *PeSPX-MFS2* expression in various tissues (leaf, stem, and root tissues). Error bars in (**A**,**B**) indicate SD (*n* = 3). The asterisks in (**A**,**B**) show a significant difference compared to the control using the unpaired Student’s *t* test (* *p* < 0.05).

**Figure 6 plants-12-01496-f006:**
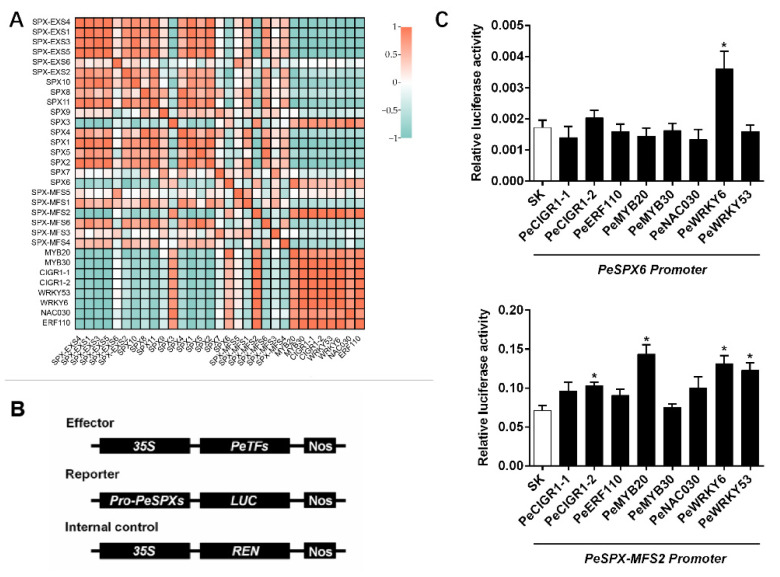
Validation of the regulatory networks of *PeSPX6* and *PeSPX-MFS2*. (**A**) Pearson’s correlation analysis between the expression levels of *PeSPX6* and *PeSPX-MFS2* and transcription factors induced by low phosphorus. (**B**) Schematic diagrams of the effector and reporter plasmids used for transient expression analysis. (**C**) Dual-luciferase analysis of the effects of potential transcription factors on LUC activity driven by the *PeSPX6* and *PeSPX-MFS2* promoter. The asterisk shows a significant difference compared to the control using the unpaired Student’s *t* test (* *p* < 0.05).

**Table 1 plants-12-01496-t001:** Description of *Phyllostachys edulis* SPX-domain-containing proteins family genes.

Gene ID	Gene Name	Location	ORF (aa)	CDS (bp)	MW (KDa)	Pi	Strand
PH02Gene06029	SPX1	chr21:3680047-3682784	315	948	35.4	5.44	-
PH02Gene33177	SPX2	chr15:98773223-98776656	316	951	35.76	5.62	+
PH02Gene04435	SPX3	chr15:32188589-32189520	252	759	28.59	5.61	-
PH02Gene50182	SPX4	chr21:57199806-57200798	254	765	28.87	5.67	-
PH02Gene21885	SPX5	chr4:8401540-8402568	257	774	29.53	5.25	+
PH02Gene10585	SPX6	chr10:6035976-6037101	245	738	27.97	5.13	+
PH02Gene17759	SPX7	chr5:30261388-30263702	260	783	29.04	9.3	-
PH02Gene49111	SPX8	chr20:37686660-37690217	283	852	31.58	5.37	-
PH02Gene12037	SPX9	chr17:86005776-86009978	277	834	31.45	5.04	-
PH02Gene39430	SPX10	chr6:41492472-41497330	295	888	33.06	5.14	+
PH02Gene13927	SPX11	chr8:59290518-59295287	293	882	32.67	5.26	-
PH02Gene14463	SPX-MFS1	chr8:3039865-3044869	701	2106	77.83	7.5	+
PH02Gene13036	SPX-MFS2	chr6:7519110-7524176	699	2100	77.49	8.43	+
PH02Gene00584	SPX-MFS3	chr24:57835339-57863679	692	2079	77.42	5.86	+
PH02Gene01924	SPX-MFS4	chr23:16516769-16531212	695	2088	77.67	6.14	-
PH02Gene01487	SPX-MFS5	chr3:82291560-82301443	686	2061	76.68	8.43	+
PH02Gene03341	SPX-MFS6	chr17:22492959-22496359	696	2091	77.88	6.65	-
PH02Gene41866	SPX-EXS1	chr6:67258648-67262438	741	2226	84.07	8.89	+
PH02Gene41868	SPX-EXS2	chr6:67340589-67344713	757	2274	86.42	8.98	+
PH02Gene10542	SPX-EXS3	chr8:35123167-35127993	856	2571	97.55	8.74	-
PH02Gene19658	SPX-EXS4	chr14:106783817-106788584	804	2415	92.16	9.25	-
PH02Gene33784	SPX-EXS5	chr3:100920611-100925663	824	2475	93.53	9.28	-
PH02Gene05859	SPX-EXS6	chr17:2650733-2655194	789	2370	89.51	9.4	+

## Data Availability

Data is contained within the article or Appendix A.

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
