# Peer review of "Genome-Wide Identification of SPX Family Genes and Functional Characterization of PeSPX6 and PeSPX-MFS2 in Response to Low Phosphorus in Phyllostachys edulis"

_plants, 2023, doi:10.3390/plants12071496_

Round 1

Reviewer 1 Report

I have read it very carefully. This MS related to genomic reserach and the whole genome of SPX in P. edulis identified and characterized. In addition, the chromosomal mapping, phylogenetic relationships, and conserved motifs of SPX genes were analyzed. Then, the expression levels of SPX members exposed to low Pi were analyzed using RNA sequencing (RNA-seq) and quantitative real-time  PCR (qRT-PCR). Furthermore, several transcriptional factors were identified that 65 modulated the expression of PeSPX18 and PeSPX21 in P. edulis.

Hovewer,  it is difficult to understand the menaing of many words,  for example, what the meaning of SPX protein and etc. Please add abbreviations paragraph.

In the Fig 3 should be indicated main source.

In addition, the literature survey needs improvement and please expand it. Additional related papers can be found in this Journal, as well as in Cells, Photosynthesis Research, Photosynthetica, Functional Plant Biology and etc., Many and very good papers has been published in these Journals on this subject. Other ways, your  MS will be relatively low interest to our readers.

Author Response

Response to Reviewer 1 Comments

I have read it very carefully. This MS related to genomic reserach and the whole genome of SPX in P. edulis identified and characterized. In addition, the chromosomal mapping, phylogenetic relationships, and conserved motifs of SPX genes were analyzed. Then, the expression levels of SPX members exposed to low Pi were analyzed using RNA sequencing (RNA-seq) and quantitative real-time PCR (qRT-PCR). Furthermore, several transcriptional factors were identified that 65 modulated the expression of PeSPX18 and PeSPX21 in P. edulis.

***Response: Thank you very much for your time and efforts in reviewing our manuscript, we appreciate a lot for your positive identification on our works.

Comment 1: It is difficult to understand the meaning of many words, for example, what the meaning of SPX protein and etc. Please add abbreviations paragraph.

***Response: Thank you very much for your excellent suggestion. We have added a paragraph of abbreviations at the end of the main text of the revised manuscript.

Comment 2: In the Fig 3 should be indicated main source.

***Response: The relevant information has been clarified in Fig 3, as well as detailed in the M&M section.

Comment 3: In addition, the literature survey needs improvement and please expand it. Additional related papers can be found in this Journal, as well as in Cells, Photosynthesis Research, Photosynthetica, Functional Plant Biology and etc., Many and very good papers has been published in these Journals on this subject. Other ways, your MS will be relatively low interest to our readers.

***Response: We sincerely appreciate this valuable comment. We have checked the literature carefully and cited more references (reference 12, 21-30, 39) on this topic related to this study in the revised version of the manuscript.

此外,毛竹是中国亚热带地区分布最广泛的种。种植面积4.677万公顷,占我国竹林总面积的800%。亚热带地区土壤磷稀少。研究表明,低磷环境会影响竹子的生理生化水平和基因转录调控。SPX基因家族在拟南芥、玉米、大豆等植物中应对低P胁迫的能力已被验证(相关文献已在文稿中引用),但尚未在竹子中进行研究。重要的是,在P. edulis基因组中没有发现SPX-RING亚家族。这证明了SPX家族在不同植物物种中的复杂性。基于上述原因,我们认为我们的手稿可以为研究毛竹响应低磷胁迫的分子机制提供依据,这应该引起许多从事经济森林和低磷响应的研究者的兴趣。

Reviewer 2 Report

The study described the genome-wide identification and expression analyses of SPX family genes in Phyllostachys edulis. A total of 25 SPXs were found in the genome, and expression analysis showed that PeSPX18 and PeSPX21 were upregulated by phosphate starvation and were potentially regulated by several transcription factors. This study could provide some preliminary data for further functional analyses.

Some comments:

1.      As mentioned in the introduction, SPX proteins can be classified into four subfamilies based on the presence of additional domains. So the name of SPXs shown in Table 1 and other figures should be named according to their different subfamilies; the nomenclature should follow those in model plants, such as Arabidopsis and Rice.

2.      The Supplemental Tables were not mentioned in results or discussion section.

3.      In Fig. 3B, SPX7 and SPX25 were classified into SPX-EXS subfamily. But there were no EXS domain in these two proteins. So they should not be the classical SPX-EXS, and they should be classified into other group.

4.      The heatmap shown in Figure 4D does not indicate detailed treatment, and their fold changes seem the same. The value of bar should be adjusted to differentiate the fold changes of these genes.

5.      Fig. 5A, the kind of tissue should be mentioned.

6.      Fig. 6A, the detailed information of these transcription factors, such as gene ID, should be mentioned.

7.      The construction of vectors used for transient luciferase assay should be described in details, such as the primers used for construction, the vector name, etc.

8.      The name of promoters shown in Figure 6 (C) is not correct.

Author Response

Response to Reviewer 2 Comments

The study described the genome-wide identification and expression analyses of SPX family genes in Phyllostachys edulis. A total of 25 SPXs were found in the genome, and expression analysis showed that PeSPX18 and PeSPX21 were upregulated by phosphate starvation and were potentially regulated by several transcription factors. This study could provide some preliminary data for further functional analyses.

***Response: Thank you very much for your time and efforts in reviewing our manuscript, we appreciate a lot for your positive identification on our works.

Comment 1: As mentioned in the introduction, SPX proteins can be classified into four subfamilies based on the presence of additional domains. So the name of SPXs shown in Table 1 and other figures should be named according to their different subfamilies; the nomenclature should follow those in model plants, such as Arabidopsis and Rice.

***Response: Agreed. We have renamed the SPX genes, and modified them accordingly throughout the manuscript.

Comment 2: The Supplemental Tables were not mentioned in results or discussion section.

***Response: Thank you very much for your reminding! The corresponding supplementary tables have been assigned at appropriate places.

Comment 3: In Fig. 3B, SPX7 and SPX25 were classified into SPX-EXS subfamily. But there were no EXS domain in these two proteins. So they should not be the classical SPX-EXS, and they should be classified into other group.

***Response: We apologize for this mistake. The SMART and CDD analysis of SPX genes were performed again, the results showed that these two genes only contained EXS super family, without classical EXS domain, and did not change under low phosphorus stress. Thus, we speculated that this gene did not belong to the SPX gene family and removed it from the manuscript. Accordingly, this information has been modified in the manuscript.

Comment 4: The heatmap shown in Figure 4D does not indicate detailed treatment, and their fold changes seem the same. The value of bar should be adjusted to differentiate the fold changes of these genes.

***Response: Many thanks for your suggestion! The heatmap shown in Figure 4D has been reorganized in the revision.

Comment 5: Fig. 5A, the kind of tissue should be mentioned.

***Response: The kind of tissue have been added into the diagram notes in Figure 5A in the revision.

Comment 6: Fig. 6A, the detailed information of these transcription factors, such as gene ID, should be mentioned.

***Response: Thank you very much! We have added this information in the supplementary table 4 which is cited in the revised manuscript.

Comment 7: The construction of vectors used for transient luciferase assay should be described in details, such as the primers used for construction, the vector name, etc.

***Response: Thank you for your suggestions! We have perfected the vector construction information in supplementary Table 2.

Comment 8: The name of promoters shown in Figure 6 (C) is not correct.

Response: We are sorry for this oversight. In this revised vertion, we have revised it.
